# Impact of Physico-Chemical Properties of Cellulose Nanocrystal/Silver Nanoparticle Hybrid Suspensions on Their Biocidal and Toxicological Effects

**DOI:** 10.3390/nano11071862

**Published:** 2021-07-20

**Authors:** Dafne Musino, Julie Devcic, Cécile Lelong, Sylvie Luche, Camille Rivard, Bastien Dalzon, Gautier Landrot, Thierry Rabilloud, Isabelle Capron

**Affiliations:** 1INRAE, Institut National de Recherche Pour L’agriculture, L’alimentation et L’environnement, BIA, Biopolymères Interactions et Assemblages, 44316 Nantes, France; dafne.musino@inrae.fr; 2Laboratoire de Chimie et Biologie des Métaux, University Grenoble Alpes, CNRS, CEA, IRIG, CBM, UMR5249, 38000 Grenoble, France; devcicjulie@gmail.com (J.D.); cecile.lelong@univ-grenoble-alpes.fr (C.L.); sylvie.luche@cea.fr (S.L.); bastien.dalzon@cea.fr (B.D.); 3SOLEIL Synchrotron, L’Orme des Merisiers, Gif-sur-Yvette, 91192 Saint-Aubin, France; Camille.Rivard@Synchrotron-Soleil.Fr (C.R.); gautier.landrot@synchrotron-soleil.fr (G.L.); 4INRAE, Institut National de Recherche Pour L’agriculture, L’alimentation et L’environnement, BIA, TRANSFORM, 44316 Nantes, France

**Keywords:** CNC/AgNP hybrids, NaBH_4_/AgNO_3_ molar ratio, H_2_O_2_ redox, oxidation state, short- and long-term biocidal effect, MIC, toxicology

## Abstract

There is a demand for nanoparticles that are environmentally acceptable, but simultaneously efficient and low cost. We prepared silver nanoparticles (AgNPs) grafted on a native bio-based substrate (cellulose nanocrystals, CNCs) with high biocidal activity and no toxicological impact. AgNPs of 10 nm are nucleated on CNCs in aqueous suspension with content from 0.4 to 24.7 wt%. XANES experiments show that varying the NaBH4/AgNO3 molar ratio affects the AgNP oxidation state, while maintaining an fcc structure. AgNPs transition from 10 nm spherical NPs to 300 nm triangular-shaped AgNPrisms induced by H_2_O_2_ post-treatment. The 48 h biocidal activity of the hybrid tested on *B. Subtilis* is intensified with the increase of AgNP content irrespective of the Ag^+^/Ag_0_ ratio in AgNPs, while the AgNSphere−AgNPrism transition induces a significant reduction of biocidal activity. A very low minimum inhibitory concentration of 0.016 mg AgNP/mL is determined. A new long-term biocidal activity test (up to 168 h) proved efficiency favorable to the smaller AgNPs. Finally, it is shown that AgNPs have no impact on the phagocytic capacity of mammalian cells.

## 1. Introduction

In the last decades, silver nanoparticles (AgNPs) have emerged as one of the most efficient biocidal agents, limiting or preventing microorganisms’ growth [1,2,3,4]. AgNPs are widely used in several applications (e.g., paints [5], cosmetics [6], dental material [7], water treatment [8]) because of their high surface-to-volume ratio [9,10] and their relatively low toxicity for human health [11]. Even if the exact action mechanism of AgNPs on bacteria is still not completely understood, several authors consider that their antimicrobial effect is mainly due to the release of the Ag^+^ ions. These ions can interact with the components of the cell (e.g., thiols) [12], inducing structural and morphological changes in bacteria [10,12,13] and leading to its subsequent inactivation [13] (i.e., bacterial cell lysis).

Depending on the application (e.g., surgical sutures) [14], a long-term constant antimicrobial activity could be required and, consequently, AgNPs are added in large excess in the system. This approach could lead to an excessive instantaneous Ag^+^ release or to a persistence of unused AgNPs, which can have a negative impact on human health [15,16] and on the environment [17,18]. An efficient alternative to obtain the biocidal properties ensured by AgNPs and by most of the metallic NPs has not yet been found. Thus, the minimization of the amount of AgNPs without being detrimental to their bactericidal effect remains a challenge at this time. In this context, a good alternative may be provided by biocidal hybrid nanomaterials, where AgNPs are grafted onto the surface of a bio-based substrate, maximizing the AgNP specific surface and ensuring their good dispersion and stabilization, even without the addition of capping agents. Furthermore, the expected degradation of the bio-based substrate at the end-of-life of the material may provide the synthesis of more eco-friendly biocides. The use of polysaccharides as reducing and stabilizing agents for AgNPs, such as cellulose nanocrystals (CNCs), is a promising way [19,20,21]. CNCs are available, sustainable, and biodegradable, with high specific surfaces, low density, and low cost. They can be extracted from several renewable natural sources [22], they are well-dispersible in water with the ability to self-assemble [23], and they thus represent an excellent bio-based support to be used in hybrid nanomaterials.

Several methods have been developed to obtain the formation of AgNPs anchored on CNC surfaces, forming a hybrid nanoparticle (CNC/AgNP). As for the synthesis of isolated AgNPs, chemical reduction (e.g., by sodium borohydride [24], hydrazine [25], or dimethylformamide [26]) is one of the most efficient methods to nucleate AgNPs on CNC surface. Generally, an Ag precursor (silver nitrate, AgNO_3_) is added to the CNC suspension and the chemical reduction of the Ag^+^ ions is performed by sodium borohydride (NaBH_4_) [27,28,29], NH_4_OH [30], polydopamine [31], or hydrazine [32]. Otherwise, CNCs are themselves indicated as both supports and reducers for the generation of metallic nanoparticles [33]. In most of the studies reported in the literature, CNCs are preliminary surface-modified (e.g., periodate oxidation, polydopamine coating) to ensure the grafting of AgNPs even if it has been experimentally proven that the AgNP nucleation on CNC surface occurs thank to the CNC surface hydroxyl groups and no surface treatment is necessary [34].

The characteristics of AgNPs (e.g., size, shape, structure) are crucial for their resulting antimicrobial activity and they strongly depend on the synthesis methods. Several works [35,36,37] reported that the smallest AgNPs provide the best efficiency in Ag^+^ release as given by the lowest inhibitory concentration (MIC) to ensure a detectable biocidal activity against various bacteria strains (e.g., *Streptococcus mutants*, *Escherichia coli*, *Staphylococcus aureus*), regardless of the fact that AgNPs are stabilized on CNCs [31,38,39] or by other capping agents or stabilizers. In line with the generic biocidal activity of Ag^+^ ions, AgNPs are toxic not only for bacteria, but also for eukaryotic cells. Comparing AgNP with different sizes (i.e., 10, 50, 100 nm), Kim et al. [40] proved that the strongest toxicological effect against eukaryotic cells was provided by the smallest NPs as the kinetics of AgNP dissolution and the consequent Ag^+^ release mechanism are driven by the NP surface-to-volume ratio.

The CNCs can also raise toxicological concerns. Their toxicity profile, including pulmonary, oral, dermal, and cytotoxicity, has been described [41,42], indicating that pulmonary toxicity is the main pathway to be considered [43]. Indeed, CNCs have been shown to induce pulmonary inflammation, but of a much lower amplitude than the one caused by true pulmonary toxicants such as asbestos [44].

Along with size, AgNP shape and structural composition can also affect the resulting antimicrobial activity [45]. The morphological characteristics of AgNPs can be modified performing specific post-treatment on synthetized AgNPs. To this aim, several authors [24,28,46,47] propose a hydrogen peroxide (H_2_O_2_) redox post-treatment as H_2_O_2_ has oxidation/reduction capabilities thanks to the autocatalytic decomposition on the AgNP surface. Indeed, the addition of H_2_O_2_ could induce a transition from spherical AgNPs to bigger Ag nanoprisms with a triangular shape (AgNPrisms) [47,48], which display a localized surface plasmon resonance peak (LSPR) in the UV spectrum. The literature concerning the biocidal activity of AgNPrisms is quite controversial. Some studies [45,49] comparing truncated triangular AgNPrisms, spherical AgNPs, and rod-shaped NPs conclude that AgNPrisms allow the most efficient inhibition of *E. coli* growth, attributing the strong biocidal activity of AgNPrisms to the relevant role of their sharp edges and vertices. In contrast, Raza et al. [50] show that AgNPrisms with an average edge length of 150 nm display a less intense antibacterial activity than those associated with spherical AgNPs with an average diameter of 15–50 nm, highlighting the fact that small spherical AgNPs could also penetrate more easily into the bacteria. The AgNP surface charge [51], surface coating [52], and solubility [53] can influence the resulting bactericidal effect as well, as Ag^+^ release seems to be involved in the regulation of antibacterial activity [1,35,54]. A few other works [2,55] focus on the role of the AgNP oxidation state (i.e., ratio between metallic silver, Ag_0_, and ionic silver, Ag^+^, in AgNP), not clearly indicating the specific contribution of the Ag^+^ and Ag_0_ fractions in AgNPs to the final biocidal activity. Nevertheless, a deeper investigation of the dependence of biocidal activity on the basis of the characteristics of AgNPs is necessary.

In our work, we proposed a detailed study focused on the correlation between the physico-chemical and morphological characteristics of AgNPs well-nucleated on the surface of CNCs and the resulting biocidal and toxicological activity. We synthetized CNC/AgNP hybrids, controlling the size, the shape, and the amount of well-dispersed AgNPs without any preliminary modification and without the use of any additional stabilizer. For the first time, we investigate the initial NaBH_4_/AgNO_3_ molar ratio as a tuning parameter of the oxidation state of AgNPs without affecting their size, shape, and structure. Differently, H_2_O_2_ redox post-treatment allowed reaching the morphological and structural transition of primary 10 nm spherical AgNPs to 300 nm AgNPrisms. The antibacterial properties of the synthesized CNC/AgNP hybrids tested on *B. subtilis* strain reveal a performant short-term activity (48 h) and allow the determination of a low MIC value. Moreover, we propose for the first time an experimental method to detect the long-term biocidal activity (up to 168 h). Finally, the toxicological activity of hybrid suspensions carried out on mammalian cells (macrophages) is described.

## 2. Materials and Methods

### 2.1. Chemicals

Cellulose nanocrystals (CNCs) were purchased from CelluForce (Montreal, QC, Canada, product no. 2015-009). These CNCs were obtained from bleached Kraft pulp by acid hydrolysis, neutralized to sodium form, and finally spray-dried (length = 183 ± 88 nm; cross section = 6 ± 2 nm; aspect ratio = 31) [23]. Silver nitrate (AgNO_3_ ≥ 99%), sodium borohydride (NaBH_4_ ≥ 96%), and hydrogen peroxide (H_2_O_2_) were purchased from Sigma-Aldrich and used without further purification. All the aqueous suspensions and solutions were prepared using ultra-pure water.

### 2.2. Synthesis of CNC/AgNP Hybrid Suspensions and H_2_O_2_ Post-Reaction

To produce well-dispersed CNC/AgNP hybrid suspensions at various AgNP contents, an experimental procedure proposed in our previous study was used [56]. Briefly, a volume of 10 mL of CNC aqueous suspension (2 g/L) was dialyzed against water for 3 days and then mixed with various amounts of AgNO_3_ aqueous solution (50 mM, from 15 to 700 μL) and with 500 μL of freshly-prepared NaBH_4_ aqueous solution (100 mM) to reduce Ag^+^ ions and to synthesize AgNPs. The NaBH_4_ aqueous solution was put in ice to minimize its decomposition. The synthesis was performed working at pH ~6. The resulting CNC/AgNP suspension was mixed for 24 h at room temperature and aluminum foil was used to protect the sample from oxidation due to room light.

The H_2_O_2_ redox post-treatment was performed by adding 160 μL of hydrogen peroxide to the primary CNC/AgNP suspension under stirring, immediately after the introduction of NaBH_4_. The addition of H_2_O_2_ initiated an exothermic reaction, leading to the formation of gas bubbles originating from the H_2_O_2_ decomposition [24].

### 2.3. UV/Vis Spectroscopy

Light-visible absorbance spectra of hybrid suspensions were recorded in the 300–900 nm range using a Mettler-Toledo UV7 spectrophotometer equipped with a 10 mm quartz cell. All the samples were diluted (1:10) by ultra-pure water, which was also used as a blank reference.

### 2.4. Atomic Absorption Spectroscopy (AAS)

The AgNP content in each hybrid suspension was determined by AAS (ICE 3300 AAS, Thermo Fisher, Waltham, MA, USA). A volume of 1 mL of sample was digested by 40 mL of water/aqua regia mixture overnight (i.e., 30% vol. aqua regia; HCl/HNO_3_: 3/1) and then analyzed. A calibration curve was obtained using a silver standard solution (1000 μg/mL, Chem-Lab NV, Zedelgem, Belgium) at different concentrations, from 0.25 to 10 ppm. For each sample, two independent measurements were performed. As the final CNC/AgNP suspension was dialyzed after the synthesis, it can be assumed that the entire Ag amount detected by AAS came from the AgNP dissolved by H_2_O/aqua regia mixture. From here on, the Ag content will thus be indicated as AgNP content, which will be expressed in mg of AgNP per g or mL of hybrid suspension (wt%).

### 2.5. Scanning Transmission Electron Microscopy (STEM)

To perform STEM acquisitions, a volume of 10 μL of diluted hybrid suspension (0.5 g/L in CNC content) was deposited on glow-discharged carbon-coated grids (200 meshes, Delta Microscopies, Microscopies, Mauressac, France) for 2 min, and the excess was then removed by Whatman filter paper. The grids were dried overnight in air and a 0.5 nm platinum layer was then deposited by an ion-sputter coater (LEICA AM ACE600, Oberkochen, Germany). Images were recorded at 10 kV with a Quattro Scanning electron microscope (Thermo Fisher Scientific, Waltham, MA, USA) using a STEM detector. ImageJ software was used to analyze the STEM images. For each sample, several images were examined and the AgNP Feret’s diameter (i.e., the greatest distance of two tangents to the contour of the measured particle) of approximately 100 AgNPs was evaluated to obtain an average value.

### 2.6. X-ray Absorption Near-Edge Structure (XANES) and Extended X-ray Absorption Fine Structure (EXAFS)

XANES measurements were performed in order to evaluate the oxidation state of AgNPs in CNC/AgNP hybrid suspension, while EXAFS made it possible to describe the AgNP bulk atomic structure considering bond length and interatomic distance. XANES and EXAFS spectra were simultaneously acquired in transmission mode at the Ag K-edge on the SAMBA beamline at the SOLEIL synchrotron (Saint Aubin, France). Energy calibration of the Si (220) monochromator was set to 25,516 eV at the first inflection point of the Ag foil XANES spectrum. To be analyzed, all the hybrid suspensions were freeze-dried and then pressed to obtain 6 mm circular pellets with an AgNP content that made it possible to reach an absorption edge jump close to 1. Subsequently, pellets were positioned on a sample rod and then placed in a liquid nitrogen bath before being introduced into the He cryostat (T = 20 K).

Two standards were considered, silver foil (Agfoil) and AgNO_3_ aqueous solution with 1 wt% glycerol (AgH_2_O). One scan was collected for each sample in transmission and continuous scan mode in the energy range of 25,250 to 27,750 eV with 5 eV/s monochromator velocity and 0.08 s/point integration time. Scans were normalized and background-subtracted using Athena [57] software. A linear combination fitting (LCF) procedure was then performed to analyze XANES spectra, in the fit range of [E_0_ − 20 eV, E_0_ + 50 eV], with E_0_ set to 25,514 eV and using Agfoil and AgH_2_O standards as components for fitting. All the component weights were forced to be positive and the relative proportions of the components were forced to add up to 100%.

EXAFS data were first background-subtracted using an autobk algorithm (Rbkb = 1, k-weight = 3), and the Fourier transform of the k^3^-weighted EXAFS spectra was then calculated over a k range of 2.5–18 Å^−1^, using a Hanning apodization window (width of the transition region window parameter = 1). The k^3^ EXAFS fitting was performed in the 2.35–7.7 Å distance range with the Artemis [57] interface to IFEFFIT using least-squares refinements. Paths used for fitting standards and samples were obtained from a metallic silver crystallographic model [58] using the FEFF6 algorithm included in the Artemis interface. The E_0_ value was fixed at 25,520 eV and only the paths with a rank higher than 7% were considered. In the fitting procedure, the amplitude reduction factor S_0_^2^ was fixed to 0.978 after being determined by fitting the first coordination sphere of the Agfoil spectrum over a range of 2.30–2.83 Å. Degeneracy of the paths, energy shift ΔE_0_, R shift ΔR, and thermal and static disorder σ^2^ were fitted for each of the selected paths for a total of 52 independent points and 19 variables. All R-factors were lower than 0.04.

### 2.7. X-ray Diffraction Spectroscopy (XRD)

A Bruker D8 Discover diffractometer was used to record XRD diffractograms of the CNC/AgNP dried samples (10 min of acquisition). Cu-Kα1 radiation (1.5405 Å) was produced in a sealed tube at 40 kV and 40 mA, parallelized using a Gobël mirror parallel optic system and then collimated to produce a 500 mm beam diameter. The data were collected in a 2θ angle range from 3° to 70°. The AgNP crystallite size (CS) was determined using Scherrer’s equation [59]:$$\mathrm{CS}=\frac{K\lambda }{\beta \cos\theta }$$
where K is the shape factor (0.9), λ is the X-ray wavelength (1.54 Å), β is the full-width at half-maximum (FWHM), and θ is the angle of the diffraction peak of the crystalline phase (Bragg’s angle). The FWHM was determined considering the AgNP characteristic peak at 2θ = 38°.

### 2.8. Ag^+^ Relesase Analysis

Conductometric measurements were performed to determine Ag^+^ release in pure water from AgNPs in hybrid suspensions. The conductivity of dialyzed CNC/AgNP hybrid suspensions was monitored at 22 °C at different times (i.e., 48, 96, 120, and 168 h) using a Metrohm 856 Conductivity Module and recorded by Tiamo^TM^ Titration Software. The concentration of Ag^+^ ions (mM) was deduced using a calibration curve obtained by the analysis of various amounts of a silver standard solution (1000 μg/mL, Chem-Lab NV, Belgium) diluted in a 2 g/L CNC suspension (i.e., Ag^+^ concentrations from 0 to 2.4 mM).

### 2.9. Biocidal Tests

For disk diffusion biocidal tests, the *B**acillus subtilis* strain was used (3610 strain, wild type, personal gift of Maria Laaberki). *B. subtilis* is a commonly used Gram-positive bacterial strain and it was selected because of its easy manipulation under L1 conditions, with minimal safety concerns. The cells were grown in LB medium: 10 g/L tryptone, 5 g/L yeast extract, and 5 g/L NaCl. A 3610 *Bacillus subtilis* culture grown overnight on liquid LB at 37 °C was diluted to A_600nm_ = 0.1 in 10 mL of fresh LB medium and incubated at 37 °C and 200 rpm until the A_600nm_ reached the exponential phase (≈0.6). A volume of 300 μL of this culture (or 10^8^ cells) was uniformly applied on the surface of an LB agar plate (22 mL LB-agar) before placing the 5 mm diameter paper disks (blotting paper, grade 703 from VWR). For each hybrid suspension, a controlled volume of 3 μL was immediately deposited on the paper disks. All the tests were performed working at neutral pH. Ultra-pure water and 70% ethanol were used as negative and positive controls, respectively. After 48 h at 30 °C, the average diameter of the inhibition zone was measured using ImageJ software. It was confirmed that the pure CNC did not display any biocidal activity (data not shown).

For the biocidal activity persistence test, the biocide-impregnated paper disks from the above experiment were recovered, layered on a fresh bacteria plate exactly as described above, and rehydrated with 3 µL of water. The plates were grown at 30 °C for 96, 120, or 168 h, and the average diameter of the inhibition zones was measured. It was estimated that 168 h represented a good compromise between the requirements for investigating a long-term biocidal activity and the feasibility of the experiment associated with the necessity of paper disk rehydration.

All experiments were performed in triplicate (three independent growth cultures) with at least two technical replicates, and the size of the paper disk was subtracted from the inhibition halo values.

### 2.10. Toxicological Tests

The effects of CNC/AgNP hybrids on mammalian cells were assessed using a murine macrophage cell line model. To do this, J774A1 cells were grown in Dulbecco modified Eagle medium (DMEM) supplemented with 10% (*v*/*v*) fetal calf serum. Cells were seeded at 500,000 cells/mL, allowed to grow for 24 h, and then treated with the hybrid suspensions for an additional 24 h before the readout. Cell viability was assessed in a flow cytometry mode using propidium iodide exclusion. Phagocytic ability was assessed by internalization of fluorescent latex beads (diameter: 1 µm) in flow cytometry mode. Interleukine-6 (IL-6) and tumor necrosis factor alpha (TNF) production were assessed using a commercial kit (CBA from Becton Dickinson, Le Pont de Clai, France). For each sample, three independent measurements were performed.

## 3. Results and Discussion

### 3.1. Variation of AgNP Content in CNC/AgNP Hybrids

A range of CNC/AgNP hybrids was produced by adding various amounts of Ag precursor (AgNO_3_) to a CNC aqueous suspension, varying the AgNP content from 0.4 wt% to 24.7 wt%, as reported in Table 1. The reduction of Ag^+^ ions by NaBH_4_ led to an immediate color change of the suspension from translucent to yellow, indicating the formation of AgNPs. As shown in the inset of Figure 1a, the color suspension became darker with increasing AgNP content. The formation of AgNPs resulted from the aggregation of monomeric Ag particles obtained by a reduction to zero-valence Ag atoms [28], and was confirmed by the presence of a dominant in-plane absorption peak at $\lambda_{\max}$ ~400 nm in the UV/vis spectra (Figure 1a) whose intensity increased with AgNP content. The STEM images of dried CNC/AgNP hybrids (Figure 1b) clearly showed well-dispersed AgNPs attached to the CNC surface. Such a result highlights the fact that CNC is an excellent substrate for AgNP nucleation, growth, and stabilization, without any CNC surface modification required [34]. Independently from the AgNP content, the average AgNP diameter measured from electron microscopic images was in the order of 10 nm (Appendix A). It turned out that the increase of precursor concentration, AgNO_3_, only determined the number of grafted AgNPs on CNC without affecting their size. After AgNP formation, characteristic Ag peaks were detectable in the XRD patterns for all the CNC/AgNP hybrid suspensions (Appendix A). The peak at 38.1° usually associated with the (111) lattice plane of face-centered cubic (fcc) silver was recorded (JCPDS Card No. 89-3722). Other characteristic Ag peaks with weaker intensity were detected at 44.1° and 64.2°, corresponding to (200) and (220) crystalline planes, respectively, thus confirming the isotropic nature of the crystals [28]. The average crystal size (CS) was estimated to be equal to 3.3 ± 0.3 nm.

It is known that AgNPs are composed of metallic Ag_0_, but they could also contain a certain Ag^+^ ion content [2,55], thus influencing the final AgNP oxidation state. The oxidation state of AgNPs in the hybrid system was investigated here by the analysis of XANES spectra. As reported in Appendix A, the XANES data referring to the hybrid sample were compared to the XANES spectra of pure silver references. The analysis of these data using the Athena software based on the LCF procedure allowed precisely quantifying the Ag^+^ and Ag_0_ fractions in AgNPs.

The resulting XANES spectra of all the samples are plotted in Appendix A, the R-factor and the Chi-square values of the LCF fits are reported in Appendix A, and the outcomes of the analysis are summarized in Table 1. It was not possible to process all the samples during the synchrotron session (limited beamtime); however, we assume that, for AgNP content below 3%, the fractions of Ag0 were at least 90%. As shown in Figure 1c, the proportion of Ag_0_ sharply decreased from around 90 to 30% with the increase of the initial AgNO_3_ concentration (i.e., increase of the AgNP content) and the subsequent lowering of the NaBH_4_/AgNO_3_ molar ratio. Even if the NaBH_4_ reducing agent was always added in excess with respect to the AgNO_3_, allowing the formation of well-defined spherical nanoparticles (i.e., NaBH_4_/AgNO_3_ molar ratio always higher than 1.5), the reduction process of the Ag^+^ ions led to a variation of the AgNP oxidation state when the AgNP content was increased, suggesting that the nucleation and/or growth of AgNPs was strongly affected by the reducing conditions. It could be concluded that AgNPs were formed by both Ag_0_ and Ag^+^ fractions and that the final Ag_0_/Ag^+^ ratio was influenced by the initial NaBH_4_/AgNO_3_ molar ratio (Figure 1c). The presence of silver oxide, Ag_2_O, in AgNP was excluded using an Ag_2_O reference for the fitting of XANES spectra where the amount of silver oxide represented less than 1%.

The EXAFS Fourier transform spectra (Appendix A) of the CNC/AgNP hybrid suspensions with increasing AgNP content were fitted with the crystallographic structure of metallic silver. For each CNC/AgNP sample, the fitted and experimental spectra were similar to each other (Appendix A). R-factors were all lower than 0.040 (Appendix A), which reflected reasonable fits [60]. This indicated that the structure of each CNC/AgNP hybrid agreed with the structure of metallic silver. The shift in R space obtained from the fits for each atomic shell is systematically negligible. The results showed that the interatomic distances in the CNC/AgNP hybrids did not significantly change in comparison with the metallic silver distances and that the space group of the AgNPs still corresponded to the fcc silver structure, as indicated by XRD. Thus, the crystal structural organization was not affected by the AgNP content.

Finally, CNC/AgNP hybrids were plunged in water and their conductivity was measured over time to estimate the Ag^+^ kinetic release from AgNPs. An increase in the Ag^+^ concentration during the first 48 h was recorded (Appendix A), in agreement with the work of Zhang et al. [61]. After 120–168 h, all the samples reached an equilibrium state with a constant Ag^+^ concentration, and the final concentration value increased with increasing AgNP content (Figure 1d). Such a result could be used to evaluate the amount of AgNP required for an Ag^+^ release desired at equilibrium conditions.

### 3.2. Morphological Variation from Spherical AgNPs to AgNPrisms by H_2_O_2_ Redox Post-Treatment

As shown in STEM images (Figure 2), the H_2_O_2_ redox action turned the primary ~10 nm AgNPs nucleated on CNC surfaces into bigger AgNPrisms. Our group recently investigated variations of morphological and physico-chemical properties of AgNP as a function of the H_2_O_2_/AgNP mass ratio [56]. In the present work, we briefly summarized the main characteristics of the CNC/AgNP hybrid suspensions where an H_2_O_2_ redox post-treatment was performed. These hybrids, referred to as CNC/AgNP_H_2_O_2_, were obtained by mixing hybrids bearing 8.7, 12.5, 18.6, and 24.7 wt% AgNP with 160 µL of H_2_O_2_. The H_2_O_2_/AgNP mass ratio, α, was equal to 0.27, 0.20, 0.12, and 0.09, respectively. After the H_2_O_2_ treatment and dialysis of the suspension, the AgNP_H_2_O_2_ content was checked by AAS and found to be equal to 6.9, 9.3, 17.1, and 24.1 wt%, respectively. We can thus assume that roughly the same amount of AgNP is measured after the post-treatment.

The transition from ~10 nm spherical AgNPs (Figure 1b) to AgNPrisms with an average diameter of 150–300 nm (Figure 2) was obtained only when the critical α value of 0.20 was exceeded. Below this threshold value, the AgNPs_H_2_O_2_ maintained their spherical shape, with a size of approximately 15 nm. Moreover, the AgNP–AgNPrism transition was associated with variations in the oxidation state. Indeed, the formation of AgNPrisms corresponded to an increase in the Ag_0_ content from 57–65% up to 100%, while the Ag_0_ fraction remained between 30 and 50% for the 15 nm spherical AgNPs_H_2_O_2_. The characteristics of the hybrids with and without the H_2_O_2_ redox post-treatment are reported in Table 2.

### 3.3. Short-Term Biocidal Activity: Impact of AgNP Content and Oxidation State

Biocidal tests were performed on *B. subtilis* for all hybrids bearing primary spherical AgNPs. Bacillus subtilis was chosen as a model organism for the following reasons: (i) it is easy and safe to handle (Biosafety level 1), (ii) it is a good model of Gram+ bacteria, and (iii) it has been demonstrated to be of similar sensitivity to metal-based nanobiocides to pathogenic microorganisms. This has been demonstrated both on copper [62] and silver [63] nanoparticles. The inhibition halo diameter measured at 48 h increased with the amount of AgNP (Figure 3a and Appendix A). According to the known bactericidal mechanism, the biocidal effect is directly linked to the released Ag^+^ ions that interact with bacteria. Such a release mechanism might be simulated by the Ag^+^ release when the hybrids are maintained in water, as previously mentioned. Figure 3a shows the Ag^+^ ion concentration at 48 h in the aqueous medium that was monitored by conductimetry. The amount of Ag^+^ released increased with the AgNP content in the hybrid, showing a slope change of around 10 wt% AgNP. This non-linear evolution of the Ag^+^ ion release was also reflected in the non-linear enlargement of the inhibition halo. Thus, the Ag^+^ release estimated by conductimetry was compared to the Ag^+^ and Ag_0_ fractions in AgNPs determined by XANES (Appendix A). The Ag^+^ fraction in the AgNPs increased with the AgNP content, as observed for the [Ag^+^] released in water. However, conductometric measurements also showed that an Ag^+^ release was detectable even when AgNPs were mainly composed of Ag_0_ (at low AgNP contents), suggesting that both metallic and complexed ionic silver can provide the Ag^+^ release and, consequently, an antimicrobial activity.

To better understand the impact of the AgNP oxidation state on the detected antimicrobial effect, we compared the biocidal activity at 48 h of hybrid suspensions containing the same AgNP content, but composed of a different Ag^+^ fraction. It should be recalled that all AgNPs had a spherical shape with an average diameter of approximately 10 nm. Firstly, three hybrids with increasing AgNP content were selected: H1, H2, and H3 at 3, 8.7, and 18.6 wt% AgNP, respectively, where the fraction of Ag^+^ varied (Table 3). The sample containing the higher concentration in Ag^+^ (H3) was then diluted twice (H3/2) and six times (H3/6) to maintain the Ag^+^ fraction constant, but decreasing the AgNP contents to 8.7 and 3.0 wt% of AgNP, respectively (i.e., as in H1 and H2). For suspensions at the lowest AgNP content, H1 and H3/6, the width of the inhibition halos was 2.9 and 4.2 mm for 8.7 and 68% Ag^+^, respectively (Table 3), which are in the error bars. Similarly, at a greater amount of AgNPs, the inhibition halos of H2 and H3/2 hybrids reached very close values of 7.1 mm and 7.4 mm for 35 and 68% Ag^+^, respectively (Appendix A). Because of the slight variation of the inhibition halos of the samples within the error bar, a clear difference between the Ag_0_ and the Ag^+^ role in the biocidal activity could not be deduced, and the Ag_0_/Ag^+^ ratio of AgNP did not seem to affect the resulting antimicrobial effect. As the AgNP oxidation state is governed by the amount of chemical reducer with respect to the metallic precursor (i.e., decrease in Ag_0_ content with the decrease of the NaBH_4_/AgNO_3_ molar ratio), the very low amount of reducer can be used to obtain well-dispersed AgNP in hybrids and, as a result, CNC/AgNP hybrids with highly efficient biocidal properties.

The evaluation of the short-term biocidal activity allowed estimating the minimum inhibitory concentration (MIC). According to the results of the diffusion tests reported in Figure 3a, the first inhibition halo was detected for the CNC/AgNP hybrid at 1.6 wt% AgNP (i.e., 0.016 mg AgNP/mL of hybrid). This MIC value is results one of the lowest values reported in literature for systems similar to the one presented in our work [31,38,64]. However, it is quite challenging to properly compare the MIC values as they strongly depend on the effective volume deposited on the diffusion disk during the biocidal activity test. Thus, the evaluation of the AgNP amount deposited on the diffusion disk can represent a more accurate parameter to determine the efficiency of the biocide. In our case, 3 μL of hybrid suspension at 1.6 wt% AgNP was used to impregnate the paper disk for the test, thus only 0.048 μg of AgNP was effectively deposited onto it. To the best of our knowledge, this is the lowest AgNP content that makes it possible to obtain a well-detectable biocidal effect compared with those proposed in other studies, as reported in Appendix A, proving the antimicrobial efficiency our hybrid system linked to the good stabilization and dispersion of AgNPs on CNC surface.

### 3.4. Short-Term Biocidal Activity: Impact of AgNP Size–Shape Variation by H_2_O_2_

A new independent test was performed to compare the antibacterial action at 48 h of CNC/AgNP hybrid suspensions with various amounts of AgNP (i.e., 8.7, 12.5, 18.6, and 24.7 wt%) with and without a fixed amount of H_2_O_2_ for redox post-treatment (i.e., α equal to 0.27, 0.20, 0.12, and 0.09, respectively); Figure 3b. It should be recalled that this type of H_2_O_2_ reaction induced the transition from spherical AgNPs with an average diameter of approximately 10 nm to 150–300 nm triangular Ag objects only for hybrids at α values higher than 0.20. The characteristics of the investigated samples are reported in Table 2.

For the CNC/AgNP hybrid at the lowest amount of AgNP, 8.7 wt%, the size of the inhibition halo for the reference decreased from 6.7 ± 0.1 mm to 1.5 ± 0.1 mm once treated with H_2_O_2_ (Figure 3b). It can be assumed that the variation of the biocidal activity could be mainly attributed to the AgNP size–shape transition to AgNPrisms, with an Ag_0_ fraction in AgNPs of between 65% and 95%. This agreed with the general concept that, the smaller the NP, the stronger the biocidal effect results will be [2,13,36,37], because the Ag^+^ release is a phenomenon occurring at the NP surface [61]. On the other hand, the AgNP size–shape transition did not occur when α was equal to 0.12 and 0.09. Thus, the AgNPs_H_2_O_2_ maintained their morphological characteristics and the Ag_0_ fraction of the primary AgNPs as before the H_2_O_2_ treatment, thus providing an equivalent biocidal activity. Accordingly, conductometric measurements showed that the release of Ag^+^ at 48 h for the larger AgNPrisms was significantly lower than that provided by spherical AgNP_H_2_O_2_ of about 10 nm (inset Figure 3b), which showed an Ag^+^ release equivalent to that of primary AgNPs. These results suggested that the accessibility and, consequently, the specific surface of AgNPs is the main factor affecting AgNP biocidal activity, and not their internal structure at 48 h.

### 3.5. Long-Term Biocidal Activity of CNC/AgNP Hybrids

A new protocol was designed to detect the biocidal activity over several days (see Material and Methods section). We focused on hybrids at 8.7 and 24.7 wt% AgNP with and without H_2_O_2_ redox post-treatment (see Table 2). As indicated in the previous sections, all these hybrids displayed an antimicrobial activity at 48 h (Figure 3b). At 96 h, a decrease of biocidal activity of all the hybrids was detected after the transfer of the hybrid-impregnated disk to a fresh bacteria plate (Figure 4). In particular, no biocidal effect was detected for the hybrids containing AgNPrisms of 150–300 nm (6.9 wt% AgNP_H_2_O_2_), suggesting that even the long-term activity was mainly governed by the AgNP size, even if it was still linked to the oxidative dissolution and, consequently, Ag^+^ ion release. After 96 h, the Ag^+^ release seemed to reach an equilibrium state for all the samples according to the conductimetry data (Appendix A). At this stage, the biocide-impregnated paper disk was rehydrated with 3 µL of H_2_O and incubated again against bacteria. At 120 h and 168 h total incubation time, no biocidal effect was detected for the hybrid suspension characterized by AgNPrisms, while an antibacterial activity was still observed for the other samples containing smaller spherical AgNPs. This experimental evidence once again underlined how the 10 nm nano-spherical shape ensured a more efficient short- and long-term biocidal effect compared with the 300 nm AgNPrisms.

Furthermore, at 168 h, the hybrid suspension at a higher AgNP content with and without the H_2_O_2_ redox post-treatment showed a more transparent inhibition zone around the disk compared with that displayed by the hybrid sample at lower AgNP content characterized by the presence of AgNPrisms. (Appendix A). Such a difference could be explained by considering not only the inhibition of the bacterial growth around the disk, but also the elimination of all the residual bacterial colonies, showing a consistent biocidal effect (i.e., complete bacterial lysis or inhibition). Moreover, the inhibition halo of the CNC/AgNP_H_2_O_2_ was slightly wider than the one of the same sample without the H_2_O_2_ post-treatment. This experimental evidence seemed to suggest that, even if a size–shape transition did not occur, the H_2_O_2_ post-treatment favored the oxidative dissolution of AgNPs_H_2_O_2_, and thus the Ag^+^ release, thus leading to a clearer biocidal activity compared with the primary AgNPs at the same total silver content, especially in medium- (96 h) and long-term (120–168 h) experiments.

To summarize, the results of our biocidal diffusion tests clearly showed that the tuning of the morphological and physico-chemical properties of the AgNPs in the CNC/AgNP hybrids made it possible to control the resulting short- and/or long-term biocidal activity. In particular, it was demonstrated that the AgNP oxidation state did not affect the Ag^+^ release associated with the antimicrobial activity, whereas the morphological transition from ~10 nm AgNPs to ~300 nm AgNPrisms induced the inhibition of the biocidal properties of the hybrid system from 48 h up to 168 h, examining the critical role of the specific surface of AgNP. Moreover, it was shown that, when the H_2_O_2_ post-treatment did not induce a morphological variation of AgNPs (i.e., no transition to AgNPrisms), their biocidal properties were enhanced, making it possible to obtain a complete bacterial lysis around the diffusion disk. It shall be underlined that, in complex biological media such as those used for the biocidal activity tests, numerous organic and inorganic Ag ligands are present, which may limit the concentration of free Ag^+^ ions at any time. The biological macromolecules bind silver with very high affinity owing to the cooperative nature of their interaction with silver ion [65], as demonstrated for example on polyamine acids [66]. Thus, even in a context where most of the liberated Ag^+^ ions are complexed by the media components, bacterial macromolecules will still be able to displace the equilibrium toward their silver-loaded forms, and eventually inactivate them and lead to the biocidal activity.

### 3.6. Impact of the Variation of AgNP Content on Macrophage Viability and Function

By definition, a biocide is not specific for a given set of species. In that humans may come into contact with the biocide during its manufacture or use, it is interesting to examine its toxicity to mammalian cells. In this context, one of the most relevant cell types to test the toxicity of a particular biocide is represented by the macrophages, as these cells are in charge of removing foreign particulate material from the body. Firstly, we examined the toxicity of CNC/AgNP hybrids containing various amounts of AgNP (from 3 to 24.7 wt%) on macrophages, directly adding them to the macrophage culture medium in various concentrations. The concentrations of CNC/AgNP suspensions in the culture medium are reported in Table 4. The results of these experiments show a toxicity only for the higher AgNP contents, at 5% and 10% of the hybrids (Figure 5), i.e., for 50% cell death, as observed for the points at approximately 20 µg/mL Ag concentration, as reported in Table 4). A moderate toxicity was observed at lower AgNP concentrations (20% cell death at approximately 10 µg/mL Ag concentration, as reported in Table 4). Such a curvilinear response curve, with no toxicity observed at low doses and a relatively sudden onset of toxicity, has been observed on the same cell type with free AgNPs [67,68].

Pure CNCs were used as a control and did not show toxicity, even at high concentrations (200 µg/mL). As shown by the comparison of the different biocides, the cytotoxic effect was relatively independent of the density of grafting, and depended instead on the AgNP concentration. The toxicity shown by the CNC/AgNP hybrids was in line with the results published in the literature [43]. Large AgNPs (diameter: 60 nm; PVP-coated) were less toxic for the same cells than the hybrids, with only 20% cell death at 20 µg/mL Ag, [67] showing once again that the critical determinant for toxicity is the size of the AgNPs. This size dependency of toxicity has been described for free AgNPs on macrophages [69,70] and on other mammalian cells [71]. A comparative analysis of our results with those of the literature showed that, at equal AgNP size of 10 nm, the toxicity of the hybrids on macrophages (EC50 15.5–20 µg Ag/mL) was similar to that of free, citrate-coated nanoparticles (16.9 µg Ag/mL) on fibroblasts, [71], but lower than that of silica-supported uncoated nanoparticles (10 µg Ag/mL) on macrophages [72].

This means that it is quite difficult to know whether the observed toxicity arose from a pure nanoparticles effect or from an intracellular dissolution of the nanoparticles and liberation of toxic Ag^+^ ions. The latter hypothesis is favored in the literature [35] and it is rooted in from the fact that nanomaterials are confined in the lysosomes in animal cells, and thus cannot exert general cellular toxicity, and in the fact that intracellular AgNP dissolution was documented [73]. In line with these observations, intracellular release of Ag^+^ ions from AgNPs was observed in the same cell type (macrophages) here used for toxicological testing [67,68]. Moreover, studies on the same macrophage cells with silver ion showed an EC20 at 1.6 µg/mL [67], i.e., close to the EC20 close to 4–5 µg/mL shown here with the hybrids with small nanoparticles. Thus, overall, it is likely that the toxicity of the hybrids for animal cells arises mainly from the release of toxic silver ion.

We also explored the functional effects of the hybrids at non-toxic concentrations. We first investigated the effects on the phagocytic capacity of the cells, i.e., their ability to ingest pathogens such as bacteria. The results reported in Figure 6a showed a major effect of the hybrids on the proportion of phagocytic cells. However, this effect was mainly due to the CNCs and did not depend on the AgNP concentration. The absence of the effect of AgNP on the phagocytic capacity at non-toxic concentrations has been previously described [67]. In addition to the proportion of phagocytic cells, we also investigated if the cells that were positive for phagocytosis under different conditions were able to internalize the same number of fluorescent beads. A slight, but significant effect was observed (Figure 6b), even if it remained constant regardless of the material used, and was thus provoked by the presence of CNCs. This inhibitory effect of high concentrations of CNCs on phagocytosis is consistent with previous studies in the literature [74].

Finally, we tested the capability of the CNC/AgNP hybrids to induce an inflammatory reaction in macrophages by measuring the secretion of the pro-inflammatory cytokines TNF-α and interleukin IL-6 (Figure 7a,b, respectively). Concerning TNF-α, a synergistic effect of CNCs and AgNP was detected, making it difficult to clearly define their respective roles in the inflammatory process. On the other hand, very low production of pro-inflammatory cytokines was observed for IL-6 A, probably induced by high concentrations of CNCs alone and not by the presence of the grafted AgNPs, as previously observed for long silver nanowires [75] or for pure CNC [74,76]. Overall, the presence of the hybrids did not show an increased toxicity for macrophages compared with free AgNPs, and their functional effects on macrophages were mostly triggered by the CNC moiety, independently of the AgNP content.

## 4. Conclusions

In this study, we focused on the link between the physico-chemical, morphological, and structural characteristics of AgNPs in CNC/AgNP hybrid suspensions and their biocidal and toxicological properties.

Firstly, the proposed simple process including the chemical reduction of Ag^+^ ions using NaBH_4_ allows the nucleation and growth of well-dispersed AgNPs on the CNC surface. No CNC surface pre-treatment was required to obtain AgNPs highly dispersible in aqueous medium. The AgNP content was varied between 0.4 and 24.7 wt%, with a narrow size-distribution of about 10 nm, whereas the amount of metallic silver (Ag_0_) estimated by XANES decreased with the reduction of the initial NaBH_4_/AgNO_3_ molar ratio from 68 to 1.5.

An H_2_O_2_ redox post-treatment made it possible to adjust the morphological properties of the primary AgNPs. When the H_2_O_2_/AgNP mass ratio (α) was higher than 0.20, an efficient AgNP size–shape transition from ~10 nm AgNPs particles to ~300 nm AgNPrisms occurred, which was associated with an increase in Ag_0_ content up to 97%, while preserving the initial fcc structure.

The physico-chemical, morphological, and structural properties of the AgNPs in CNC/AgNP hybrids were associated with their antibacterial effect on *B*. *subtilis*. At 48 h, biocidal activity increased with the increase of the silver content in CNC/AgNP hybrids, but the oxidation state of AgNPs in hybrids (Ag^+^/Ag_0_ ratio) did not trigger the short-term biocidal effect. In particular, AgNPs synthesized at a very low NaBH_4_/AgNO_3_ molar ratio also provided an efficient antimicrobial activity. Such a result opens the way to an eco-friendlier AgNP synthesis with a noticeable decreased amount of chemical reducer. The evaluation of the short-term biocidal activity allowed us to determine an MIC of 0.016 mg AgNP/mL of hybrid. Such an MIC corresponded to an effective AgNP amount deposited on the diffusion disk of 0.048 μg of AgNP. To the best of our knowledge, this is the lowest AgNP content that makes it possible to obtain a well-detectable biocidal effect. These results proved that the control of AgNP properties and distribution on a substrate allow strongly reducing the silver content without being detrimental to the biocidal activity, and recovering a bio-degradable cellulosic substrate after use of the material, thus reducing the environmental impact.

No biocidal activity was detected at 48 h when the H_2_O_2_ post-treatment induced a transition to the larger 300 nm AgNPrisms. This result agrees with the lower Ag^+^ released with the increase in particle size (i.e., reduction of the NP specific surface area).

We also proposed a specific method to experimentally detect a long-term biocidal activity of up to 168 h. As for the tests at 48 h, smaller spherical AgNPs provided the most efficient long-term biocidal activity, while larger AgNPrisms (150–300 nm) inhibited the antibacterial effect. However, when the H_2_O_2_ redox post-treatment did not induce large particles, the oxidation-reduction steps seemed to favor the Ag^+^ release. To the best of our knowledge, this is the first experimental study where the long-term antibacterial activity is measured. Furthermore, the precise comparative study of well-dispersed spherical AgNPs and AgNPrisms led to conclusive results on the inefficiency of AgNPrisms compared with smaller Ag nanospheres.

Finally, the toxicological results showed no specific impact of AgNP when it was grafted on the CNC substrate. As CNCs were previously shown to solely promote pulmonary toxicity and only at high concentrations, it is advisable to take precautions to limit their aerosolization. In this respect, the fact that the hybrid synthesis takes place via a wet route and that the end product is a suspension that can be directly used in applications is a positive aspect.

To conclude, it is by nature impossible to obtain a biocidal product that is both highly efficient and totally harmless for humans and the environment. The present results offer a very complete safe-by-design reflection on the preparation of a biocidal product. The present study allows a highly controlled amount of AgNPs to be used with high efficiency in anticipation of the release of a cellulose-safe product into the environment after use. This precisely designed bio-based biocidal material opens the way for many applications such as medical instruments or devices, wastewater treatment, food packaging and processing, or paints with biocidal properties, where the AgNP content and properties can be optimized as a function of the desired antibacterial effect.

## Figures and Tables

**Figure 1 nanomaterials-11-01862-f001:**
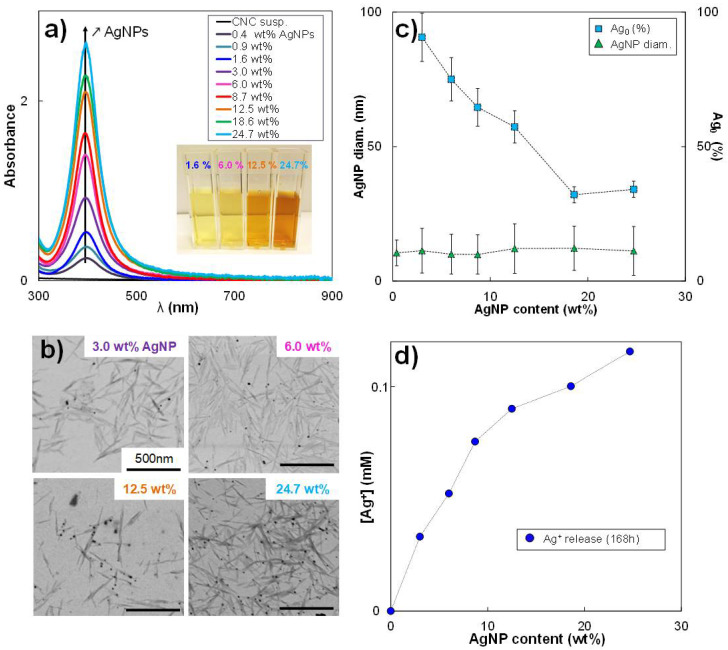
(**a**) UV/vis spectra and (**b**) STEM images of CNC/AgNP hybrids at various AgNP contents; (**c**) Evolution of AgNP average diameter and Ag_0_ fraction in AgNPs as a function of the AgNP content in the hybrid suspension; (**d**) Ag^+^ release plateau value at 168 h in pure water.

**Figure 2 nanomaterials-11-01862-f002:**
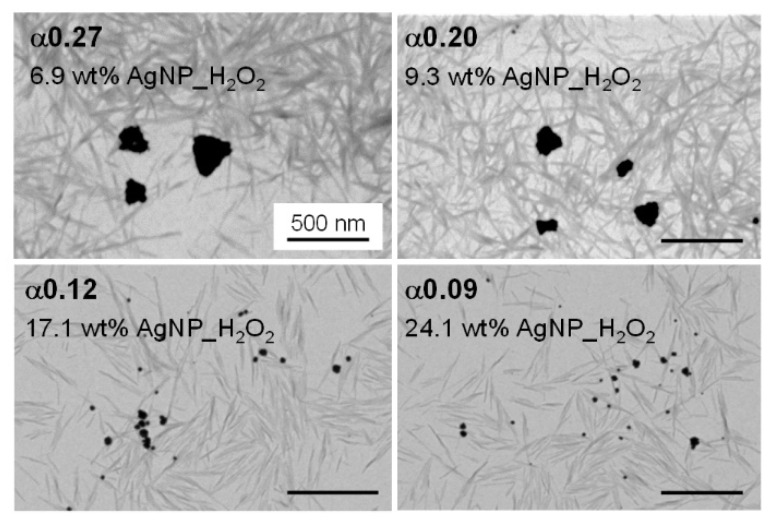
STEM images of CNC/AgNP hybrids at various α values. The scale bar is 500 nm.

**Figure 3 nanomaterials-11-01862-f003:**
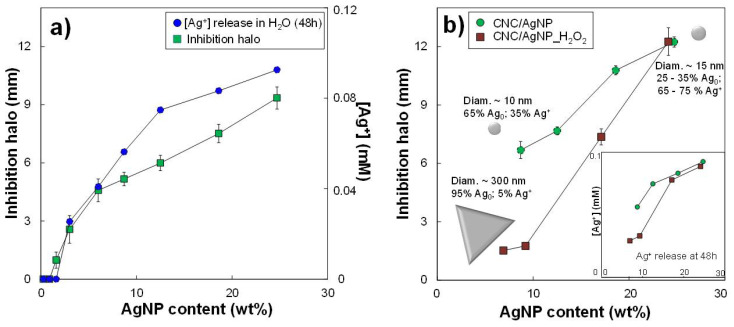
(**a**) Variation of the inhibition halo diameter and of the Ag^+^ ion concentration released in water measured by conductimetry as a function of the AgNP content in the CNC/AgNP hybrid suspensions. All AgNPs have an average diameter of ~10 nm. (**b**) Comparison between inhibition halos of CNC/AgNP at different AgNP contents with and without H_2_O_2_ redox post-treatment. Inset: Evolution of Ag^+^ release in water at 48 h measured by conductimetry.

**Figure 4 nanomaterials-11-01862-f004:**
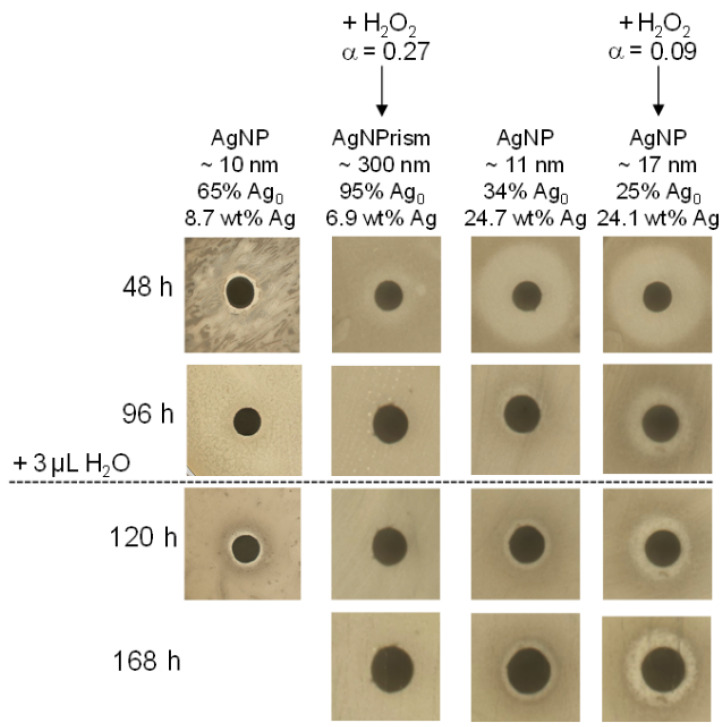
Evolution at various incubation times (48, 96, 120, 168 h) of inhibition halos of CNC/AgNP hybrids at two different initial AgNP contents (8.7 and 24.7 wt% Ag) with and without H_2_O_2_ redox post-treatment.

**Figure 5 nanomaterials-11-01862-f005:**
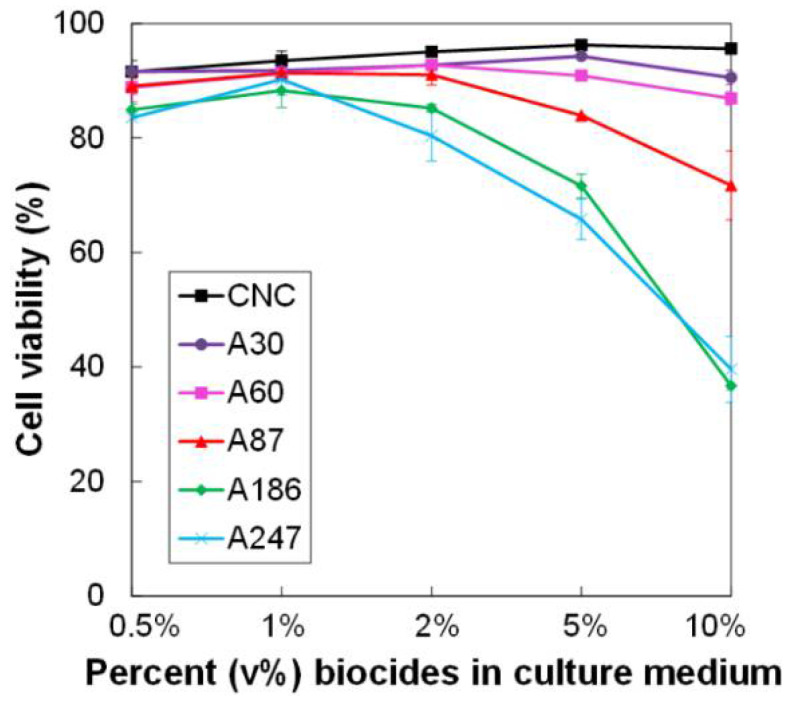
Assessment of toxicity of biocides on J774A1. Dose-dependent survival curve for cells treated for 24 h with various concentrations of hybrids.

**Figure 6 nanomaterials-11-01862-f006:**
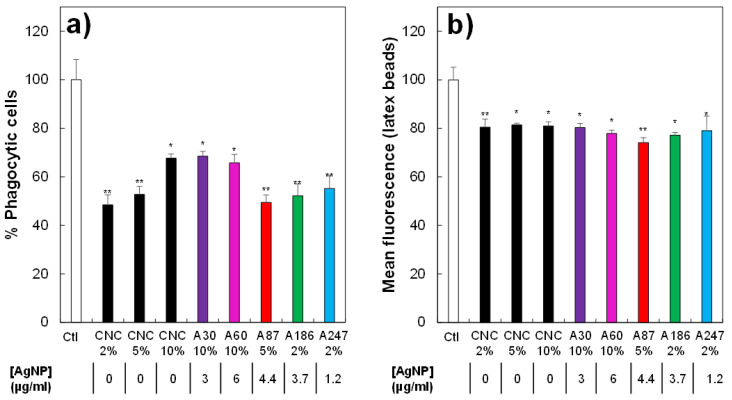
Effects of biocides on the phagocytic capacity of J774A1. (**a**) Proportion of phagocytic cells (in the viable cell population only) for control cells or cells treated for 24 h with different biocide concentrations; (**b**) mean fluorescence of phagocytic cells (in the viable cell population only) for control cells or cells treated for 24 h with different concentrations of biocides. Selected biocide concentration = before reaching lethal dose 20. Symbols indicate the statistical significance (Student’s *t*-test): * *p* < 0.05; ** *p* < 0.01.

**Figure 7 nanomaterials-11-01862-f007:**
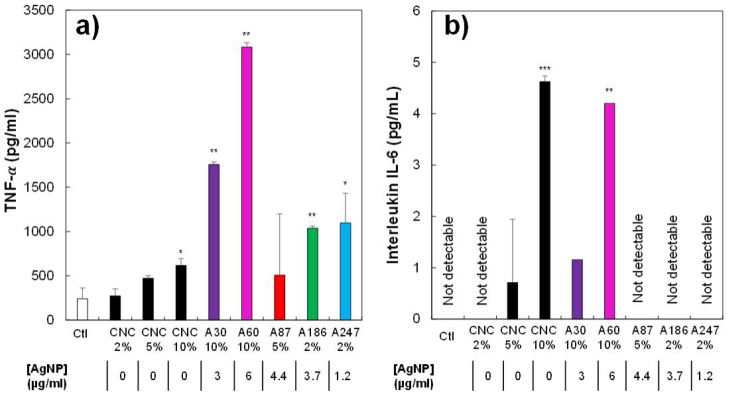
Effects of biocides on the inflammatory response of J774A1. (**a**) TNF-α production of cells only (white bar) or cells treated for 24 h with different concentrations of biocides; (**b**) IL-6 production of cells only or cells treated for 24 h with different concentrations of biocides. Selected biocide concentration = before reaching lethal dose 20. Symbols indicate the statistical significance (Student’s *t*-test): * *p* < 0.05; ** *p* < 0.01; *** *p* < 0.001.

**Table 1 nanomaterials-11-01862-t001:** Characteristics of cellulose nanocrystal (CNC)/AgNP hybrids at different AgNP contents.

AgNO_3_ Vol. (μL, 50 mM)	NaBH_4_/AgNO_3_ Molar Ratio	AgNP Content (mg Ag/g Hybrid, wt%) ^1^	Avg. Feret’s Diam. (nm) ^2^	CS (nm) ^3^	Ag_0_ (%) ^4^
15	67.7	0.4 ± 0.03	10.5 ± 4.8	3.5	-
30	33.4	0.9 ± 0.04	-	-	-
60	16.7	1.6 ± 0.02	-	-	-
110	9.1	3.0 ± 0.03	11.3 ± 8.3	-	91 ± 9
160	6.3	6.0 ± 0.01	10.0 ± 7.4	3.7	75 ± 8
240	4.2	8.7 ± 0.05	9.9 ± 7.3	3.1	65 ± 7
330	3.0	12.5 ± 0.16	12.1 ± 9.2	3.2	57 ± 6
550	1.8	18.6 ± 0.14	12.2 ± 8.2	-	32 ± 3
700	1.5	24.7 ± 0.34	11.2 ± 9.1	3.0	34 ± 3

- Not measured, ^1^ by atomic absorption spectroscopy (AAS), ^2^ by scanning transmission electron microscopy (STEM) image analysis, ^3^ crystal size determined by X-ray diffraction (XRD), ^4^ by X-ray absorption near-edge structure (XANES); the standard error is established as 10% of the measured value.

**Table 2 nanomaterials-11-01862-t002:** Characteristics of hybrids with and without H_2_O_2_ post-treatment.

AgNP Content (wt%)	H_2_O_2_ Vol. (µL)	H_2_O_2_/AgNP Mass Ratio, α	NP Shape/Size ^1^ (nm)	Ag_0_ (%) ^2^
8.7	0	0	Nanospheres (~10)	65 ± 7
12.5	0	0	Nanospheres (~10)	57 ± 6
18.6	0	0	Nanospheres (~10)	32 ± 3
24.7	0	0	Nanospheres (~10)	34 ± 3
6.9 (AgNP_H_2_O_2_)	160	0.27	Nanoprisms (~150–300)	95 ± 10
9.3 (AgNP_H_2_O_2_)	160	0.20	Nanoprisms (~150–300)	97 ± 10
17.1 (AgNP_H_2_O_2_)	160	0.12	Nanospheres (~15)	50 ± 10
24.1 (AgNP_H_2_O_2_)	160	0.09	Nanospheres (~15)	29 ± 3

^1^ by STEM image analysis; ^2^ by XANES; the standard error is established as 10% of the measured value.

**Table 3 nanomaterials-11-01862-t003:** Characteristics of hybrid samples used in biocidal tests to discriminate the antibacterial activity of Ag^+^ and Ag^+^ fractions in AgNPs.

Hybrid (Code)	AgNP Content (mg/g of Hybrid), %	Ag^+^ (%) by XANES	Inhibition Halo (mm)
H1	3.0	9	2.9 ± 1.4
H2	8.7	35	7.1 ± 0.2
H3	18.6	68	10.0 ± 0.9
H3/2 (eq. H2)	9.3	68	7.4 ± 0.8
H3/6 (eq. H1)	3.1	68	4.2 ± 1.1

**Table 4 nanomaterials-11-01862-t004:** Concentrations of AgNP for various hybrid volume fractions analyzed for toxicity on J774A1.

Sample Code	Relative AgNP Content in Hybrid (wt%)	AgNP in Culture Medium (µg/mL)				
		0.5 v%	1 v%	2 v%	5 v%	10 v%
A30	3	0.2	0.3	0.6	1.5	3
A60	6	0.3	0.6	1.2	3	6
A87	8.7	0.4	0.9	1.7	4.4	8.7
A186	18.6	0.9	1.9	3.7	9.3	18.6
A247	24.7	1.2	2.5	4.9	12.4	24.7

## Data Availability

Additional results are in supporting information, no other publicly archived datasets analyzed or generated during the study.

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
