# Peer review of "Impact of Physico-Chemical Properties of Cellulose Nanocrystal/Silver Nanoparticle Hybrid Suspensions on Their Biocidal and Toxicological Effects"

_nanomaterials, 2021, doi:10.3390/nano11071862_

Round 1

Reviewer 1 Report

The work presented in the submitted manuscript is of the scope and interest of the readers of this journal. The paper could be accepted for publication after making additional experiments with silver ions obtained directly form aqueous solution of silver nitrate, all to eliminate any antimicrobial effect from reducing agents, and also to propose cytotoxicity effect coming from the CNC/AgNP hybrid. Detailed suggestions are listed below:

-              The abstract should present only the most significant data, not all                    of them;

-              In the Introduction section:

               *             line 49, bacteria are type of microorganisms, therefore no                                need to mention them separately;

               *             line 50 and following, is written: " several fields (e.g.,                                       paints,[5] cosmetics,[6] dental material,[7] water                                               treatment[8])", the fields need to be described more                                         precisely as the types of fields as medical, daily life and                                   other...;

               *             " the components of the cell membrane (e.g., thiols)[12],"                               (line 54) it should be clearly mentioned that they are                                       bacterial membranes as authors focus on antibacterial                                     biopotency of AgNps;

-              Materials and Methods:

               *             line 166, is written: " Transmission electron microscopy                                   (STEM)." It should be SEM, please correct in any place of                                 the manuscript;

-              In the Results and Discussion:

               *             lines 276 and below, in Table 1 presenting "Characteristics                               of CNC/AgNP hybrids at different AgNP contents", why                                   the few first AgNO3 vol results, 15, 30 and 60, there is no                                 any measurements provided for Ag (0)?...;

               *             lines 282-284, in the legend for Figure 1 is written: "(a)                                     UV-Vis spectra; and (b) STEM images of CNC/AgNP                                         hybrids at various AgNP 282 contents; (c) Evolution of                                     AgNP average diameter and Ag0 fraction in AgNPs as a                                  function of the 283 AgNP content in the hybrid                                                suspension;    (d) Ag+ release plateau value at 168 h in                                    pure water",

                            it should be provided the method used to obtain results                                  presented in (c) and (d) parts; in (b) should be mention                                    "TEM";

               *             in Fig. 1 (c) a Y axis presenting AgNP diam (nm) the scale                               should be between 0 and 15 nm to clearly present some                                 diversity, it should be corrected;

*          In lines 377-380, the authors have written: “Since the AgNP oxidation              state is governed by the amount of chemical reducer with respect to              the metallic precursor (i.e., decrease in Ag0 content with the                            decrease  of the NaBH4/AgNO3 molar ratio), the very low amount of              reducer can be used to obtain well-dispersed AgNP in hybrids and,                 as a result, CNC/AgNP hybrids with highly efficient biocidal                             properties. “

To avoid any influence of even trace of NaBH4 and H2O2, it should be used a control with silver ions obtained directly from AgNO3 and with two  concentrations: 1) used exactly as it was calculated for the total amount of silver in AgNP (one experiment) and 2) the maximum Ag+ calculated as released from AgNP (second experiment). Then the results would be crystal clear and the authors could claim that their MIC is the lowest for this type of silver nanoparticles (if the additional experiments confirm it).

*             In cytotoxicity study presented in  Fig. 5-7 and Table 4, it should be                 not only CNC and CNC/AgNP hybrids tested but also silver ions                     obtained directly from AgNO3 to present the  investigation of the                  possible mechanism of  CNC/AgNP hybrids toxicity….

Author Response

Comments and Suggestions for Authors

The work presented in the submitted manuscript is of the scope and interest of the readers of this journal. The paper could be accepted for publication after making additional experiments with silver ions obtained directly form aqueous solution of silver nitrate, all to eliminate any antimicrobial effect from reducing agents, and also to propose cytotoxicity effect coming from the CNC/AgNP hybrid. Detailed suggestions are listed below:

Reply: We thank Reviewer 1 for his/her positive appreciation of our work. The amendments brought to the revised manuscript to address the requested modifications are detailed below:

*The abstract should present only the most significant data, not all of them;

Reply: The abstract has been shortened

- In the Introduction section:

*line 49, bacteria are type of microorganisms, therefore no need to mention them separately;

 Reply: done

*line 50 and following, is written: " several fields (e.g., paints,[5] cosmetics,[6] dental material,[7] water treatment[8])", the fields need to be described more precisely as the types of fields as medical, daily life and other...;

Reply: To comply with the reviewer’s remark, “fields” have been replaced by “applications” which, we agree, is more correct regarding the level of detail in the citations

* " the components of the cell membrane (e.g., thiols)[12],"(line 54) it should be clearly mentioned that they are bacterial membranes as authors focus on antibacterial  biopotency of AgNps;

Reply: Thank you for pointing out this problem, which originates indeed from a mistake of ours. Indeed silver ions can interact not only with components of the cell wall and cell membrane, but also with intracellular components. As we have tested the toxicity on both bacteria and animal cells, we prefer to keep the generic word “cell”

- Materials and Methods:

* line 166, is written: " Transmission electron microscopy (STEM)." It should be SEM, please correct in any place of the manuscript;

Reply: It is not real TEM but transmission electron microscopy obtained from a SEM instrument, ans thus Scanning Transmission Electron Microscopy. The S from Scanning has been forgotten. We thank the reviewer for the remark; it is now added in the text.

- In the Results and Discussion:

*lines 276 and below, in Table 1 presenting "Characteristics of CNC/AgNP hybrids at different AgNP contents", why the few first AgNO3 vol results, 15, 30 and 60, there is no  any measurements provided for Ag (0)?...;

Reply: At 3% of AgNP in the hybrid, the Ag0 content was already at 91% while decreasing with the increase of the AgNP content. Thus, since it was not possible to process all the samples during the synchrotron session (limited beamtime), we assumed that at the lowest AgNP fractions, the Ag0 was at least 90% and we focused on the Ag0 variation associated to the increase of the AgNP content. A sentence has been added in the text lines 422-424.

 *lines 282-284, in the legend for Figure 1 is written: "(a) UV-Vis spectra; and (b) STEM images of CNC/AgNP hybrids at various AgNP 282 contents; (c) Evolution of AgNP average diameter and Ag0 fraction in AgNPs as a  function of the 283 AgNP content in the hybrid  suspension;(d) Ag+ release plateau value at 168 h in pure water",

it should be provided the method used to obtain results presented in (c) and (d) parts; in (b) should be mention "TEM";

Reply: In b, STEM is correct

In c, it corresponds to the values given in table 1 were is specified "by STEM images analysis"

In d, Ag+ release analysis was carried out by conductometric measurement as described in the MATERIALS and METHODS part. To make it clearer, the paragraph has been changed from "Conductometry" to "Ag+ release analysis".

*in Fig. 1 (c) a Y axis presenting AgNP diam (nm) the scale should be between 0 and 15 nm to clearly present some diversity, it should be corrected;

Reply: If so the figure becomes very complicate with very large error bars. The real values are in table 1, it is then possible to see that the diameter doesn’t change whereas the Ag0 content is very different. This is not visible anymore if smaller scale is used, so we prefer to keep the figure as presented in the initial version.

*In lines 377-380, the authors have written: “Since the AgNP oxidation state is governed by the amount of chemical reducer with respect to  the metallic precursor (i.e., decrease in Ag0 content with the decrease  of the NaBH4/AgNO3 molar ratio), the very low amount of reducer can be used to obtain well-dispersed AgNP in hybrids and, as a result, CNC/AgNP hybrids with highly efficient biocidal properties. “

To avoid any influence of even trace of NaBH4 and H2O2, it should be used a control with silver ions obtained directly from AgNO3 and with two  concentrations: 1) used exactly as it was calculated for the total amount of silver in AgNP (one experiment) and 2) the maximum Ag+ calculated as released from AgNP (second experiment). Then the results would be crystal clear and the authors could claim that their MIC is the lowest for this type of silver nanoparticles (if the additional experiments confirm it).

Reply: We think that the reviewer’s concerns are largely addressed by the experiments described in Figure 3A, Figure S6 and Table 3. As the concentration of NaBH4 used to make the various hybrids at various silver concentrations is constant, the diluted high-silver hybrids (H3/2 and H3/6) contain by definition a lower NaBH4 concentration (and a higher ionic silver concentration) than their lower silver counterparts (resp. H2 and H1). However, the biocidal effects are identical between preparations where the only common parameter is the amount of total silver, thereby showing that this parameter is the critical one and not others such as NaBH4 (which is the way quickly hydrolyzed in aqueous media to form borate) or ionic silver concentration. The same reasoning also holds true for H2O2, where i) no additional biocidal effect is detected in samples prepared with H2O2 compared to H2O2-free samples, and ii) the biocidal effect does not correlate with the H2O2 concentration but with the size of the particles

*In cytotoxicity study presented in  Fig. 5-7 and Table 4, it should be  not only CNC and CNC/AgNP hybrids tested but also silver ions obtained directly from AgNO3 to present the  investigation of the  possible mechanism of  CNC/AgNP hybrids toxicity….

Reply: These experiments on the toxicity of silver ion on the very same macrophage line have been published in ref 67 (Dalzon et al 2019). We now remind the main results of this reference and compare them to the ones obtained with the new hybrids.

Reviewer 2 Report

This manuscript describes the manufacture of silver nanoparticles and cellulose-AgNP hybrids and their biocidal and toxicological activity.  The manuscript is well written, and the data was well presented and analysed.  I think the manuscript should be accepted for publication if some relatively minor criticism can be defended.

(1) Why was B.subtilis chosen as the bacterium in the biocidal assays? It is not a clinical issue, nor is problematic with respect to antibiotic resistance.  It would have been more useful to conduct these assays with Pseudomonas aeruginosa or Acinetobacter baumanii, or some other problematic strain. A more comprehensive assessment of target susceptibility should be conducted.  

(2) B. subtilis is in fact a spore former.  With the assay system used by the researchers, there only vegetatively cells would be able to be inhibited. It would be interesting to know if this hybrid system can function in liquid culture, since one would think that sporulated cultures would not be as susceptible. …which brings me to my other point; is this effect bacteriostatic or bacteriocidal? 

(3) When examining cytokines produced from cells, it’s very difficult to make any conclusions from single timepoint experiments.  This is because cytokines are temporally regulated and released at different times.  The Figure 7 shows that there is some TNF-alpha that is being released as a result of the silver. Therefore, it would be important to measure expression at several timepoints over 24h.  Why were these two cytokines chosen.  I would say that IL-1beta is also very important in the induction of inflammation and activation of acute phase response. This should be included as well.

Author Response

This manuscript describes the manufacture of silver nanoparticles and cellulose-AgNP hybrids and their biocidal and toxicological activity.  The manuscript is well written, and the data was well presented and analysed.  I think the manuscript should be accepted for publication if some relatively minor criticism can be defended.

Reply: We thank Reviewer 2 for his/her positive appreciation of our work. The amendments brought to the revised manuscript to address the requested modifications are detailed below:

(1) Why was B.subtilis chosen as the bacterium in the biocidal assays? It is not a clinical issue, nor is problematic with respect to antibiotic resistance.  It would have been more useful to conduct these assays with Pseudomonas aeruginosa or Acinetobacter baumanii, or some other problematic strain. A more comprehensive assessment of target susceptibility should be conducted.  

Reply

Previous papers have shown that B. subtilis offers a similar sensitivity profile to metal-based nanobiocides than pathogenic microorganisms. This has been demonstrated both on copper (Usman et al. http://dx.doi.org/10.2147/IJN.S50837) and silver (Gopinath et al. http://dx.doi.org/10.1016/j.colsurfb.2012.03.023 ) nanoparticles. This is not surprising as B. subtilis is a soil microorganism and thus usually in contact with heavy metals. Thus, B. subtilis is an easy to handle (biosafety level 1) model organism (as opposed to pathogenic microporganisms that are much more difficult to handle), and is generally used as a model for Gram+ bacteria, a family that comprises a wealth of pathogenic microorganisms (e.g. Staphylococci and Streptococci). The two references mentioned above have now been cited in the revised manuscript (lines 346-350).

(2) B. subtilis is in fact a spore former.  With the assay system used by the researchers, there only vegetatively cells would be able to be inhibited. It would be interesting to know if this hybrid system can function in liquid culture, since one would think that sporulated cultures would not be as susceptible. …which brings me to my other point; is this effect bacteriostatic or bacteriocidal? 

 Reply

For a first assessment of the biocidal activity, we have preferred to resort to classical, and well-accepted disk tests, rather than to liquid cultures, which are much more cumbersome to handle in order to give comparative results. Being able to perform quickly comparisons on different materials was indeed a critical feature for our studies. Regarding the specific issue on spores, it is highly unlikely that such biocides will have any action on spores. However, spores must germinate and proliferate to become a real problem, and at this stage the biocides will be efficient again.

Regarding the question on bacteriostatic vs bacteriocidal, there is a consensus in the literature toward a  bacteriocidal effect of silver ( Nisar et al. doi: 10.1007/s00775-019-01717-7, Ruparelia et al. doi: 10.1016/j.actbio.2007.11.006), when a zone disk inhibition test is used.

(3) When examining cytokines produced from cells, it’s very difficult to make any conclusions from single timepoint experiments.  This is because cytokines are temporally regulated and released at different times.  The Figure 7 shows that there is some TNF-alpha that is being released as a result of the silver. Therefore, it would be important to measure expression at several timepoints over 24h.  Why were these two cytokines chosen.  I would say that IL-1beta is also very important in the induction of inflammation and activation of acute phase response. This should be included as well.

Reply

In the cell culture system that we use, IL1 beta requires a priming with LPS to induce its expression, which precludes in turn the possibility of measuring relevantly TNF and IL6, as the LPS priming effect on IL6 and TNF secretion will be far greater than the effect of any nanoparticular material. Furthermore, TNF and IL6 are key players in the pro-inflammatory response, and their measurement at the end of a 24 hours exposure period is of standard practice in nanotoxicology to make a raw assessment of the inflammatory potential of nanomaterials (e.g. in doi:10.1016/j.tox.2009.10.034, doi: 10.1016/j.toxlet.2015.09.025, doi:10.1371/journal.pone.0124368, doi: 10.1007/s00204-014-1210-1, doi: 10.1039/c9na00721k). As the focus of the paper is on the biocidal activity and not on the inflammatory potential, which is only a side experiment, albeit important, we think that standard measurements are sufficient to demonstrate our point. We do agree that effect persistence may be an interesting phenomenon to study, as suggested by the reviewer, but this appears to us to be beyond the scope of the present paper.

Reviewer 3 Report

In this interesting study the authors propose a new method to structure silver nanoparticles with a nanocrystal cellulose substrate. At the same time they determine the biocidal activity and the toxicological impact. The results obtained appear to be interesting and promising. Further studies will be needed to confirm these results.

Author Response

We thank Reviewer 3 for his/her positive appreciation of our work

Round 2

Reviewer 1 Report

The manuscript can be accepted for publishing after removing deleted fragments presented on side and confusing the final look at the revised version.